# Incidence and Levels of Aflatoxin M_1_ in Artisanal and Manufactured Cheese in Pernambuco State, Brazil

**DOI:** 10.3390/toxins15030182

**Published:** 2023-02-28

**Authors:** Isabela Maria de Moura Silva, Adriano Gomes da Cruz, Sher Ali, Lucas Gabriel Dionisio Freire, Luzianna Macedo Fonseca, Roice Eliana Rosim, Carlos Humberto Corassin, Rodrigo Barbosa Acioli de Oliveira, Carlos Augusto Fernandes de Oliveira

**Affiliations:** 1Department of Consumer Science, Post-graduate program in Food Science and Technology, Federal Rural University of Pernambuco (UFRPE), Pernambuco 52171-900, PE, Brazil; 2Department of Foods, Federal Institute of Education, Science and Technology of Rio de Janeiro (IFRJ), Rio de Janeiro 20270-021, RJ, Brazil; 3Department of Food Engineering, Faculty of Animal Science and Food Engineering (FZEA), University of São Paulo (USP), Pirassununga 13635-900, SP, Brazil; 4Department of Animal Husbandry, Luiz de Quiroz Higher School of Agriculture (ESALQ), University of São Paulo (USP), Piracicaba 13418-900, SP, Brazil

**Keywords:** cheese, AFM_1_, contamination, HPLC

## Abstract

Cheese is one of the most susceptible dairy foods to accumulating aflatoxins due to their high affinity to caseins. The consumption of cheese contaminated with high levels of aflatoxin M_1_ (AFM_1_) can be highly harmful to humans. The present work, based on high-performance liquid chromatography (HPLC), highlights the frequency and levels of AFM_1_ in coalho and mozzarella cheese samples (*n* = 28) from the main cheese-processing plants in Araripe Sertão and Agreste in the state of Pernambuco, Brazil. Of the evaluated cheeses, 14 samples were artisanal cheeses and the remaining 14 were industrial (manufactured) cheeses. All samples (100%) had detectable levels of AFM_1_, with concentrations ranging from 0.026 to 0.132 µg/kg. Higher levels (*p* < 0.05) of AFM_1_ were observed in artisanal mozzarella cheeses, but none of the cheese samples exceed the maximum permissible limits (MPLs) of 2.5 µg/kg established for AFM_1_ in cheese in Brazil and 0.25 µg/kg in the European countries by the European Union (EU). The high incidence of low levels of AFM_1_ found in the evaluated cheeses underscores the need for stringent control measures to prevent this mycotoxin in milk used for cheese production in the study area, with the aim of protecting public health and reducing significant economic losses for producers.

## 1. Introduction

Aflatoxins are mycotoxins produced as secondary metabolites by species of the fungal genus *Aspergillus*, mainly *A. flavus*, *A. parasiticus*, and *A. nomius*, during growth on food and feed products [1]. These fungi produce a range of toxic metabolites, but the main compounds produced under natural conditions are aflatoxins B_1_ (AFB_1_), G_1_ (AFG_1_), B_2_ (AFB_2_), and G_2_ (AFG_2_) [2,3]. All the aflatoxins are highly toxic to humans, causing various effects such as hepatotoxicity, mutagenicity, teratogenicity, immunosuppression, and carcinogenicity, among other effects [1,2,3]. When animals ingest feed contaminated with the most prominent and highest carcinogenic AFB_1_, the compound is biochemically converted in the animal’s liver into the hydroxylated aflatoxin M_1_ (AFM_1_), which is excreted in the milk and other biological fluids [4]. The occurrence of AFM_1_ in milk and milk-derived products is a major concern for public health, as the AFB_1_ and AFM_1_ metabolites were classified as group 1 (human carcinogens) by the International Agency for Research on Cancer (IARC) [5,6,7,8]. 

Aflatoxin M_1_ (AFM_1_) binds strongly to casein in milk, resulting in greater levels in dairy products that are high in protein, such as cheeses. Moreover, traditional processes used in milk product manufacture, such pasteurization or sterilization, do not destroy AFM_1_ [9]_._ Previous reports have shown that AFM_1_ levels are approximately 4 and 5 times higher in soft and hard cheeses, respectively [9,10]. Soft cheeses are habitually made by coagulating casein with acid, while hard cheeses are ripened after coagulation of the milk proteins with the rennet and culture acids. However, both types of cheeses can accumulate variable levels of AFM_1_, depending on the moisture content of the product [11], as shown in previous studies of AFM_1_ in cheeses from the states of São Paulo, Amazonas, and Santa Catarina in Brazil [11,12,13]. At present, there are no available details on AFM_1_ level in cheeses from the state of Pernambuco, Brazilian.

The main factors affecting the AFM_1_ frequencies and levels include variations in cheese manufacturing practices, changes in milk contamination, cheese ripening conditions, geographical and seasonal changes, and the analytical methods used [14]. Given the potential harm of AFM_1_, Brazil has adopted regulatory limits for this mycotoxin, with maximum permissible limits (MPL) of 0.5 µg/L for fluid milk and 2.5 µg/kg for cheese [15]. The occurrence and levels of AFM_1_ in milk products have been monitored [14] using several analytical methods, comprising the popular enzyme-linked immunosorbent assay (ELISA) [16], high-performance liquid chromatography (HPLC) alone [17], and HPLC combined with tandem mass spectrometry (MS/MS) [18]. In this context, monitoring AFM_1_ concentrations in milk and its derived products is an essential aspect of food safety efforts, aimed at protecting consumers’ health protection [19]. The aim of the present study was to evaluate the occurrence and levels of AFM_1_ in artisanal and industrial cheeses produced in the main dairy plants in the state of Pernambuco, Brazil.

## 2. Results

This study is the first to evaluate the incidence and levels of AFM_1_ in different cheeses from the main production plants in the Araripe Sertão and Agreste regions of Pernambuco state in Brazil. The occurrence and levels of AFM_1_ were determined by HPLC, as given in Table 1. The results display a high rate of AFM_1_ occurrence in all analyzed cheese samples (100%, *n* = 28). In addition, the levels of AFM_1_ were assessed, with artisanal mozzarella cheese (25%, *n* = 7) demonstrating levels ranging from 0.026 µg/kg to 0.093 µg/kg, and the manufactured mozzarella cheese (25%, *n* = 7) indicating levels ranging from 0.037 to 0.132 µg/kg. Similarly, artisanal coalho cheese (25%, *n* = 7) exhibited AFM_1_ levels ranging from 0.035 µg/kg to 0.045 µg/kg, whereas the manufactured coalho cheese (25%, *n* = 7) revealed AFM_1_ levels ranging from 0.035 µg/kg to 0.046 µg/kg. Throughout the analyzed cheese samples, the mean level of AFM_1_ for 25% of the artisanal mozzarella cheeses (*n* = 7) was 0.07 ± 0.02 μg/kg, which was significantly higher (*p* < 0.05) than the remaining artisanal or manufactured coalho cheeses.

To support the results, the levels of AFM_1_ in cheese samples were further assessed by distributing them into quartiles (Figure 1). In this analysis, the statistical comparison displayed that artisanal mozzarella cheese (AM) had a low concentration range of 0.026 µg/kg. An increased variability was found in the AFM_1_ levels, which could be compared with the remaining analyzed cheeses. Overall, AFM_1_ values for AM ranged from 0.0423 µg/kg to 0.0889 µg/kg, but some samples had concentrations below or above 0.0262 µg/kg and 0.0935 µg/kg, respectively. The median AFM_1_ concentration in AM was 0.068 µg/kg.

However, the manufactured mozzarella (MM) cheese revealed AFM_1_ values ranging from 0.037 µg/kg, which is the minimum limit, to 0.132 µg/kg, which is the maximum limit. Furthermore, 75% of the MM samples had concentrations above 0.0373 µg/kg, and the median concentration was lower compared to the other groups, although there was overlap in quartile 1 with the lower limit.

The analysis of the artisanal coalho (AC) cheese represented AFM_1_ levels at a minimum of 0.035 µg/kg and a maximum of 0.045 µg/kg, with a median of 0.04 µg/kg. In turn, the manufactured cheese (MC) revealed AFM_1_ levels ranging from 0.035 µg/kg to 0.047 µg/kg, with a median of 0.041 µg/kg. However, the MC had greater variability in the data compared to the AC (Figure 1).

## 3. Discussion

This study, based on HPLC, evaluated the incidence and levels of AFM_1_ in cheeses collected from production plants in Araripe Sertão and Agreste in the state of Pernambuco in Brazil (Table 1). The results demonstrated a high occurrence rate of AFM_1_ in all analyzed cheese samples (100%, *n* = 28) with varying levels (Table 1). None of the analyzed cheeses had AFM_1_ levels higher than the Brazilian MPL of 2.5 µg/kg [20] or the EU MPL of 0.25 µg/kg. In exception to the northeastern region, such as Pernambuco, the AFM_1_ evaluation has been shown to be present in a large number of studies conducted in various regions and states of Brazil, as described. This study, for the first time, aimed to evaluate the occurrence and levels of AFM_1_ in different types of cheeses produced in the northeastern region of Brazil. These results indicate that consuming cheeses in these regions may be potentially harmful, and proper care, evaluation, and strict quality control procedures are necessary. This is because fungi are toxic organisms that naturally cause contamination of foods and feeds. When present, these toxigenic fungi produce mycotoxins, such as AFB_1_ and its hydroxylated AFM_1_, in foods. The hydroxylated AFM_1_ can transfer from one food (milk) to another food (e.g., cheeses) during fabrication, even under strict processing and handling procedures. When the resulting cheese products contain high levels of AFM_1_, they can pose a risk to human health. To reduce this risk exposure and increase awareness in the food industry, the occurrence and levels of AFM_1_ are evaluated in various milk products, comprising different cheeses produced in Brazil.

Brazil is a major producer of cheese in the world, fabricating a wide variety of cheeses. According to the recent data reported by the Food and Agriculture Organization (FAO), around 59,543.14 tons of cheeses were produced in Brazil in 2020 [21]. The country is known for its diverse array of artisanal cheeses, many of which are considered national treasures [22]. Recently, public incentives for technological progress and commercial partnerships for cheese production have increased in Brazil [22]. This has led to growing concerns about the quality and reliability of cheese products. In the last few decades, many studies have been conducted to quantify and evaluate the occurrence level of AFM_1_ in different types of cheeses produced in various regions of Brazil. The investigation of AFM_1_ in this study is in line with previous data demonstrating the incidence of AFM_1_ in different types of cheeses produced not only in Brazil, but also in other countries (Table 2). 

In accordance with Table 2, Syllos et al. [23] were the first to evaluate the occurrence and levels of AFM_1_ in Minas cheese, mozzarella, and cheddar, which were sold in Campinas, São Paulo, Brazil. Unfortunately, their pioneering study did not distinguish AFM_1_ in the tested cheese samples, as the evaluation was conducted using thin layer chromatography [23]. Further studies have reported a higher occurrence of AFM_1_ in cheese products with varying levels. One of these studies, using HPLC, investigated three types of cheeses, namely Minas frescal (*n* = 7), canastra (*n* = 18), and Minas standard (*n* = 50). The results showed that the entire samples (*n* = 75) contained AFM_1_, with levels ranging from 0.03 to 0.18 μg/kg in Minas frescal (57.1%, *n* = 4/7), 0.02 to 1.7 μg/kg in canastra (61.1%, *n* = 11/18), and 0.02 to 6.92 μg/kg in Minas standard (82%, *n* = 41/50) cheese [24]. In this study, the average concentration of AFM_1_ was highest in the Minas standard cheese (0.62 μg/kg), followed by canasta (0.36 μg/kg), and frescal cheese (0.08 μg/kg), as given in Table 2. Furthermore, HPLC-based analysis of 25% (6 out of 24) of Minas frescal and 29.2% (7 out of 24) of Minas standard cheese were found to contain AFM_1_ at levels ranging from 0.142 to 0.118 μg/kg and 0.118 to 0.054 μg/kg, respectively [27]. This study also found that Minas frescal cheese had a higher occurrence and level of AFM_1_ (0.142–0.118 μg/kg) compared to Minas standard cheese (0.118–0.054 μg/kg) (Table 2). Amongst Minas frescal light (*n* = 20), Minas frescal (*n* = 30), and Minas standard (*n* = 8) cheese, a higher occurrence of AFM_1_ was found in 67.2% of the cheese samples, with the levels ranging from 0.01 to 0.304 μg/kg [28]. Likewise, other typical cheeses (*n* = 23) produced in Brazil, incorporating prato (*n* = 9) and parmesan ralado (*n* = 14), were analyzed and showed higher occurrence of AFM_1_. The levels of AFM_1_ in prato cheese (100%, *n* = 9/9) and parmesan ralado cheese (92.8%, *n* = 13/14) were found to be 0.02–0.54 μg/kg and 0.04–0.30 μg/kg [25]. Overall, parmesan ralado cheeses (*n* = 30), 60% (*n* = 18) showed AFM_1_ positivity at levels ranging from 50 to 690 μg/kg [29]. These parmesan cheeses were commonly sold in the metropolitan region of Rio de Janeiro, Brazil [29]. In contrast, 45.4% (*n* = 40/88) of parmesan cheese was found to contain AFM_1_, with levels ranging from 0.02 to 0.66 μg/kg [26]. 

Following additional research, a study conducted in Lebanon (Table 2) found that 67.57% (*n* = 75/111) of the locally fabricated or imported white and yellow cheese types, comprising Halloumi, Naboulsi, Feta, Baladi, and Akkawi, were contaminated with high occurrence of AFM_1_ at levels ranging from 0.00561 to 0.315 μg/kg [31]. However, the same study found that the occurrence and levels of AFM_1_ tremendously exceeded the EU permissible level of 0.25 μg/kg [31]. Additionally, data from Iran revealed that 53.8% (*n* = 194/360) of the locally produced Lighvan, Koozeh, Siahmazgi, Khiki Talesh, and Lactic cheeses were contaminated with higher levels of AFM_1_ at 0.0505–0.3087 μg/kg [33]. A study carried out in Mexico City found that 57% (*n* = 17/30) of Oaxaca cheeses tested contained AFM_1_ at an average level of 1.7 μg/kg [34]. In addition, it was found that the Oaxaca artisanal cheeses produced in Veracruz contained higher levels of AFM_1_ compared to Oaxaca cheeses fabricated in Mexico City [34]. Additional examination revealed 85% (*n* = 39/46) of various types of cheeses had higher occurrence of AFM_1_, with a mean level of 0.1977 μg/kg [36]. A report from Ethiopia found 100% (*n* = 82/82) of the cottage cheese samples tested positive for AFM_1_, with levels recorded at 5.58 ± 0.08 μg/kg [38]. In a recent study, approximately 70% (*n* = 42/60) of the cheeses produced in Serbia were found to be contaminated with AFM_1_ at levels exceeding 0.25 μg/kg [37]. The analysis of white cheese samples (*n* = 10/25) from Turkey indicated the incidence of AFM_1_ at concentrations ranging from 0.00246 to 0.035 μg/kg (mean: 0.01714 ± 0.0042 μg/kg) [32]. Similarly, a quantifiable range of AFM_1_ was detected in 29% (*n* = 8) of the Minas frescal cheese manufactured in São Paulo State, at levels ranged from 0.113 to 0.092 μg/kg [12]. However, more studies have been conducted to investigate the presence of AFM_1_ in Minas frescal cheese marketed in the northeast region of São Paulo, Brazil [27]. The incidence of this mycotoxin in foods has been shown to be dependent on several factors, including the type of food, seasonal variability, geographic location, post-harvest period, cheese-making procedures, analytical method, and cheese maturation [32,39].

The results of the coalho and mozzarella cheeses can be attributed to variations in manufacturing methods and physicochemical characteristics of each product, as described in previous studies [40,41]. The levels of AFM_1_ in cheese depend on the type of cheese, the amount of water eliminated, the curd temperature, the pH of the saturated brine, and the duration of cheese pressing. Reports suggest that AFM_1_ levels may be higher in hard cheese compared to soft cheese, due the higher amount of the milk proteins, mainly casein, to which AFM_1_ has a greater binding affinity [19]. A study in Brazil evaluated the incidence and levels of AFM_1_ in a range of milk and milk-derived products, containing several cheese samples (*n* = 57) [17]. The study found that of the analyzed products, except a high occurrence rate, only one cheese samples represented AFM_1_ at a level of 0.695 µg/kg [17]. Moreover, HPLC coupled with fluorescence-detector-based analyses of cheeses (*n* = 58), incorporating Minas frescal light, Minas frescal, Minas standard, and other type of cheese, found that 67.24% of the cheeses were contaminated with AFM_1_ (0.01–0.304 µg/kg), as given in Table 2 [28].

This work from the northeastern state of Pernambuco in Brazil detected lower levels of AFM_1_ compared to other studies conducted on almost same cheese samples in northern Brazil [11]. Similarly, the levels of AFM_1_ in artisanal and manufactured mozzarella cheeses ranged from 0.026 to 0.093 μg/kg and from 0.037 to 0.132 μg/kg, respectively, while artisanal and manufactured coalho cheeses showed 0.035–0.045 μg/kg and 0.035–0.046 μg/kg, which were comparatively lower than those detected in northern Brazil. In the northern region, especially the state of Amazonas in Brazil, 25 cheese samples (coalho, coalho de buffalo, mozzarella, mozzarella de buffalo, and Minas frescal) were analyzed, and none of the samples exceeded the Brazilian MPL of 2.5 μg/kg for AFM_1_ [11]. Moreover, in the southern region of Brazil, AFM_1_ was analyzed in Serrano artisanal cheeses at four different maturation periods (14, 21, 28, and 35 days), and it was observed that only four samples had AFM_1_ levels of 0.505, 0.875, 0.093, and 1.03 μg/kg [13]. While most of the samples had AFM_1_ levels below the Brazilian MPL of 2.5 μg/kg, some exceeded the EU MPL of 0.25 μg/kg. Although seasonal factors were not considered in the current study, the incidence and level of AFM_1_ in cheese samples analyzed can also be influenced by different time periods of cheese manufacture. The artisanal and coalho or mozzarella cheeses produced between March and May 2022 (Brazilian autumn) give AFM_1_ levels ranging from 0.026 to 0.132 µg/kg (Table 1).

It is important to note that the impact of seasonality on AFM_1_ levels in cheeses has already been described [42]. In this study, the levels of total aflatoxins and AFB_1_ in roughage, concentrate, and compound feed were low in the autumn, followed by the summer and winter, while spring had the highest level of mycotoxins [42]. This demonstrates that harvesting and proper drying of vegetable crops used for feed during the autumn may be less risky, leading to milk and cheese with lower AFM_1_ levels. Another study found a significant difference (*p* < 0.05) in the occurrence and level of AFM_1_ in traditional cheese produced in the summer and winter seasons [33]. In regard to the seasonal impact on cheese contaminated with AFM_1_, further studies have also been performed. For example, in a study of traditionally produced Egyptian cheeses over two years (2016–2017), the occurrence and concentration of AFM_1_ in karish Egyptian cheese was found to range from 1.11 to 0.632 µg/kg [35]. However, a seasonal evaluation exhibited that the karish cheeses were significantly contaminated with AFM_1_ at 1.34 µg/kg (2016) and 0.855 µg/kg (2017) only in the winter, compared to the other three seasons [35]. 

In the current study, none of the analyzed cheese samples exceeded the MPL of 2.5 µg/kg set by Brazil [20]. This limit (2.5 µg/kg) is higher compared to the MPL set by the EU of 0.25 μg/kg for dairy products [9], as well as other countries including Iran, Austria and Switzerland, and Italy, which have set MPLs of 0.25 µg/kg and 0.45 µg/kg [27]. When comparing the results, 25% of the artisanal mozzarella cheese samples showed AFM_1_ levels at 0.07 μg/kg that do not exceed any of the described MPLs. Conversely, in a study of parmesan cheese commercially available in Rio de Janeiro, Brazil, overall samples showed AFM_1_ levels below the Brazilian MPL of 2.5 µg/kg, but 26.7% samples still exceeded the EU’s MPL of 0.25 µg/kg [29]. A study in Ethiopia found that 100% (*n* = 82) of locally produced and industrialized cheese contained AFM_1_ at levels of 5.58 ± 0.08 μg/kg, exceeding the limits set by Egypt, the EU, and Morocco, and 88% of the samples even exceeded the MPL settled by the United States [38].

In a study carried out in Qatar, halloumi and kashkaval cheeses were found to have significantly higher levels of AFM_1_ compared to mozzarella, edam, cheddar, cream, and moshalal varieties [36]. The authors attributed the elevated levels of AFM_1_ in halloumi and kashkaval cheeses to factors such as maturation time, manufacturing method (industrial or artisanal), milk source, and time of production [36]. The variations in the processing of artisanal cheese were likely due to a lack of standardization, especially in the case of mozzarella cheese, which was observed during the sampling time. Factors contributing to these issues could include the use of poor-quality products, water content, incorrect pH correction, improper cooking, and cooking time of the curdled dough. The production of artisanal cheeses is a traditional method that is passed down from generation to generation and has significant economic importance, especially for small-scale producers. This production is often based on empirical methods, which can result in varying quality standards in the same production region [40,43,44].

## 4. Conclusions

According to the findings of this study, a high incidence of AFM_1_ was detected using HPLC in both artisanal and industrially produced cheeses in Pernambuco, Brazil. The levels of AFM_1_ were low and all the samples were found to be below the limit of 2.5 µg/kg established in Brazil and 0.25 µg/kg adopted in the EU. The study found significant differences between the levels of contamination of artisanal mozzarella-type cheese and manufactured coalho cheese. Based on these results, there is a need for strict quality control measure to further reduce the presence of this mycotoxin in the milk used for cheese manufacture in the studied area. Moreover, future studies should also evaluate the cheeses throughout different seasons to determine the incidence and level of AFM_1_, as seasonality can have a significant impact on cheese production in the same regions of Pernambuco, Brazil.

## 5. Materials and Methods

### 5.1. Sampling Procedures

In this study, a non-probabilistic convenience sampling was employed, which accounted for approximately 70% of the producers in the study area. Participation in the research was voluntary and not all producers agreed to take part. The collection of the samples was performed for the first time in two main cheese-processing plants in the Agreste and Araripe Sertão regions of Pernambuco state, Brazil. A total of 28 samples of mozzarella- and coalho-type cheeses, 14 of which were artisanal and the remaining 14 were industrially produced, were collected between March and May 2022. Individual samples (original package, 500 g) were stored at 4 °C prior to transport to the laboratory for immediate analytical procedures.

### 5.2. Sample Preparation

The analysis of the cheese samples was performed according to a previously in-house validated analytical method [30]. Individual samples (8 g) were taken in falcon tubes (15 mL). To each sample was added 2 g of sodium chloride (NaCl), 22 mL of methanol (CH_3_OH, Dinâmica^®^, São Paulo, SP, Brazil), and 13 mL of ultra-pure water (Mili-Q). Individual mixtures after homogenization (1 min), shaking (10 min), and centrifugation (6000 rpm × 15 min) were filtered through membrane filters (0.22 µm) in glass tubes. At this stage, 20 mL of the individual extracts were diluted in 30 mL of ultra-pure water (Mili-Q) and submitted to re-centrifugation (6000 rpm × 15 min). Now the final filtrates were subjected to passing through the immunoaffinity columns (AflaTest, Vicam, Waters, MA, USA) connected to the glass syringes and vacuum system (2–3 mL/min flow rate). The columns were washed with 20 mL ultra-pure water, and targeted AFM_1_ in the samples was eluted with 1 mL of methanol. Then, the samples were subjected to dryness on evaporation under the nitrogen flux (MultiVap-54), and the dried extracts were one-by-one reconstituted in a 1 mL solvent mixture of methanol/water (50:50, *v*/*v*). 

### 5.3. Chromatographic Analysis

The determinations of AFM_1_ were carried out on a high-performance liquid chromatography (HPLC) system (Shimadzu 10 VP, Kyoto, Japan) equipped with 10 AXL fluorescence detectors (Excitation at 360 nm and emission above 440 nm). A Kinetex C_18_ column (Phenomenex, Torrance, CA, USA) 4.6 × 150 mm, 2.6 µm particle size, and an in-line filter of 0.5 µm were used. The isocratic mobile phase consisted of methanol/water/acetonitrile (6.4:28.1:10.5, *v*/*v*/*v*) with a flow rate of 0.50 mL/min.

Calibration curves with five points were prepared by diluting AFM_1_ standard (Sigma^®^, St Louis, MO, USA) in acetonitrile (CH_3_CN) at the concentration ranges of 2.5, 5, 10, 20, 40 ng/mL. The limits of detection (LOD) and quantification (LOQ) were calculated based on the signal-to-noise ratio (S/N) of 3:1 and 10:1. The LOD and LOQ values for AFM_1_ in cheese samples under study were 0.017 and 0.055 µg/kg, respectively. All HPLC runs were carried out in triplicate and the data average values were expressed in the form of mean ± standard deviation.

### 5.4. Data Analysis

The results obtained from the trial were subjected to one-way analysis of variance (ANOVA) to determine differences among the cheese samples and the Tukey 5% test was applied. The significance level was accepted at the probability *p* < 0.05. All analyses were carried out using XLSTAT 2022 software (v.24.2.1300), with descriptive statistics and Microsoft Excel (v.14.0.4760.1000) also being used to summarize the data. Additionally, R version 4.0.5 software was used.

## Figures and Tables

**Figure 1 toxins-15-00182-f001:**
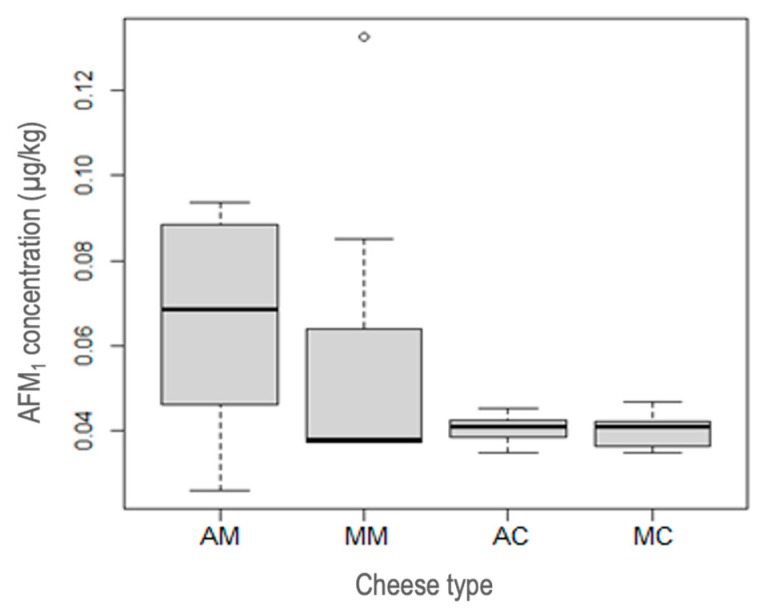
Box plots representing statistical comparison of AFM_1_ level in artisanal mozzarella (AM, *n* = 7), manufactured mozzarella (MM, *n* = 7), artisanal coalho (AC, *n* = 7), and manufactured coalho (MC, *n* = 7) cheeses. The black line inside each box, represents median.

**Table 1 toxins-15-00182-t001:** Incidence and level of aflatoxin M_1_ (AFM_1_) in cheese manufactured in Pernambuco, Brazil.

Type of Cheese	Positive Samples		Level of AFM_1_	
	*n*	%	Mean ± SD	Range (µg/kg) *
Artisanal mozzarella (*n* = 7)	7	100	0.07 ± 0.02 ^a^	0.026–0.093
Manufactured mozzarella (*n* = 7)	7	100	0.06 ± 0.03 ^a,b^	0.037–0.132
Artisanal coalho (*n* = 7)	7	100	0.04 ± 0.003 ^b^	0.035–0.045
Manufactured coalho (*n* = 7)	7	100	0.04 ± 0.004 ^c^	0.035–0.046

^a,b,c^ Means with different superscript letters differ significantly at *p* < 0.05. * Minimum and maximum levels of AFM_1_ in cheeses. None of the analyzed samples had AFM_1_ levels above the Brazilian maximum permissible limit (2.5 µg/kg) [20]. SD, standard deviation.

**Table 2 toxins-15-00182-t002:** Incidence and level of aflatoxin M_1_ (AFM_1_) in different types of cheese manufactured globally.

Sampling Year	Cheese Type	Samples (*n*)	Positive *n* (%)	AFM_1_ Level (µg/kg)	Ref.
1989–1990	Minas cheese mozzarella Cheddar	12 12 12	0 0 0	ND ND ND	[23]
1996–1998	Minas frescal Canastra Minas standard	7 18 50	4 (57.1) 11 (61.1) 41 (82)	0.03–0.18 0.02–1.70 0.02–6.92	[24]
2000–2001	Prato Parmesan ralado	9 14	9 (100) 13 (92.8)	0.02–0.54 0.04–0.30	[25]
2004	Parmesan	88	40 (45.4)	0.02–0.66	[26]
2008	Minas frescal Minas standard	24 24	6 (25) 7 (29.2)	0.142–0.118 0.118–0.054	[27]
2010	Minas frescal light Minas frescal Minas standard	20 30 8	39 (67.24)	0.01–0.304	[28]
2011	Parmesan ralado	30	18 (60)	50–690	[29]
2011–2012	NG	10	3 (30)	91–300	[30]
NG (Analysis year 2012)	Lebanese (Halloumi, Naboulsi, Feta, Baladi, Akkawi) and Imported white and yellow	111	75 (67.57)	0.00561–0.315	[31]
2014–2015	White cheese	25	10 (40)	0.00246–0.035	[32]
2015–2016	Lighvan, Koozeh, Siahmazgi, Khiki Talesh, and Lactic	360	194 (53.8)	0.0505–0.3087	[33]
2016	Oaxaca	30	17 (57)	1.7 (average)	[34]
2016–2017	Karish cheese	62	21 (39.9)	1.11–0.632 (mean)	[35]
		56	25 (44.6)		
2017	Coalho, Coalho buffalo, Mozzarella, Mozzarella buffalo, Minas frescal	25	NG	NG	[11]
2017–2018	Various	46	39 (85)	0.1977 (mean)	[36]
2018	Minas frescal	28	8 (28.6)	0.113–0.092	[12]
2018	Serrano artisanal	80	4 (5)	0.505 0.875 0.093 1.030	[13]
2019–2020	Different domestic and imported	60	42 (70)	> 0.025	[37]
2019–2020	Ethiopian Cottage cheese	82	82 (100)	5.58 ± 0.08	[38]
2022	Minas frescal	57	1 (1.7)	0.017–0.695	[17]

ND, not detected, NG, not given.

## Data Availability

The data related to this work are within the manuscript.

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
