# Peer review of "Incidence and Levels of Aflatoxin M1 in Artisanal and Manufactured Cheese in Pernambuco State, Brazil"

_toxins, 2023, doi:10.3390/toxins15030182_

Round 1
Reviewer 1 Report
The paper entitled: Incidence and levels of aflatoxin M1 in cheese manufactured in Pernambuco State, Brazil, deals with a study of levels of AFM1 in 28 cheese samples based on high-performance liquid chromatography (HPLC), concluding that all samples (100%) had detectable levels of AFM1, with concentrations ranging from 0.026 to 0.132 μ g/kg, concluding that the high incidence of low levels of AFM1 found in the evaluated cheeses reinforces the need for stringent control measures to avoid this mycotoxin in milk used for cheese manufacture in the studied area and that the low levels of the mycotoxin that were quantified showed that all of them meet the limits (2.5 μ g/kg) established in Brazil, although 7 samples (25%) exceeded limit (0.05 μ g/L) adopted in the EU. In summary, the paper is interesting in the way that compares the previous studies with the new one, albeit the number of samples is rather limited and more samples form other representative regions and types of cheese should be required for an extensive study. The methodology is not new therefore its only interest is the comparison between previous and new results.
Author Response
Reviewer comment: The paper entitled: Incidence and levels of aflatoxin M1 in cheese manufactured in Pernambuco State, Brazil, deals with a study of levels of AFM1 in 28 cheese samples based on high-performance liquid chromatography (HPLC), concluding that all samples (100%) had detectable levels of AFM1, with concentrations ranging from 0.026 to 0.132 μg/kg, concluding that the high incidence of low levels of AFM1 found in the evaluated cheeses reinforces the need for stringent control measures to avoid this mycotoxin in milk used for cheese manufacture in the studied area and that the low levels of the mycotoxin that were quantified showed that all of them meet the limits (2.5 μ g/kg) established in Brazil, although 7 samples (25%) exceeded limit (0.05 μg/L) adopted in the EU. In summary, the paper is interesting in the way that compares the previous studies with the new one, albeit the number of samples is rather limited and more samples from other representative regions and types of cheese should be required for an extensive study. The methodology is not new therefore its only interest is the comparison between previous and new results.
Authors response: According to the methodology, a non-probabilistic convenience sampling was carried out, which represented approximately 70% of the producers in the study area. It should be noted that not all producers voluntarily agreed to participate in the research. Despite this, the authors believe that the 70% participation rate was significant, particularly because it is the first study of its kind conducted in the region.

Reviewer 2 Report
The manuscript entitled “Incidence and levels of aflatoxin M1 in cheese manufactured in Pernambuco State, Brazil” is interesting as investigated the prevalence and levels of AFM1, as a major dangerous carcinogenic to humans, in artisanal and manufactured coalho and Mozzarella type cheese by using HPLC, but the following comments should be corrected as followed:
- The title should be changed as “Incidence and levels of aflatoxin M1 in artisanal and manufactured cheese in Pernambuco State, Brazil”, or “Incidence and levels of aflatoxin M1 in coalho and Mozzarella cheese in Pernambuco State, Brazil”.
- The English should be extensively edited in whole the manuscript.
- The samples number are too small to be representative for the obtained results, as well as compared with the literatures in the same area of the present work.
- In whole manuscript: change maximum permitted limit to maximum permissible limit.
- In the manuscript, HPLC was ignored as an important method of detection and quantification of AFM1, please mention in introduction, result, discussion and conclusion, as well as discuss with the other same used literatures.
- Line 6: change level to levels.
- Line 8: change type to types.
- Line 20: remove aflatoxin.
- Line 21: Add before the word first: is the “This is the first…”.
- Line 28: change cheese to cheeses.
- Line 42: remove the word five.
- Line 65: change higher to high.
- Line 68 and line 69: 0.07µg/kg and 0. 04µg/kg are different than mentioned in Table 1, please check and unify.
- Line 72: ab are conflicting, as mentioned when comparing rows and columns, this is not mentioned in this work, please revise the ststistics and only write one letter as well as revise the result and the discussion with this regard to the different types of examined cheeses.
- Line 75: the reference no. [19] is too old and since 2011 to 2023 sure several investigation so update the reference with this regard.
- Below Table 1: Describe the definition of range*.
- The median value mention in line 91 0.041 is wrong it should be 0.04 as well as line 93 the median value mentioned is 0.041 it is wrong and should be 0.040, please revise all values and correct them.
- Line 92: the word similar is wrong as they are not similar as the max level was 0.046µg/kg.
- Line 97: change IC to MC.
- Line 99: complete the sentence.
- Line 103: remove the word table 1 and rewrite the sentence again.
- The figure 1: is mismatched with the values mentioned in table1, especially MM, please correct for all values for all types of examined cheeses.
- Discussion:
· You should mention your result first then compare it with others.
· The collected compared literatures should be exclusively with the examined two types of cheeses: artisanal and manufactured coalho and Mozzarella, please collect more literatures with this regards to make discussion fair and sounds good.
· Rewrite the discussion with the previously mentioned regards.s
- Table 2:
· The comparison should be exclusively with the examined two types of cheeses: artisanal and manufactured coalho and Mozzarella.
· In the mentioned unit used ng/Kg and mg/Kg but your compared unit is µg/kg, how come? So you should harmonize between the discussed units to be fair as writing foot note below each unit to the equivalent µg/kg.
- Line 172: 250ng/Kg is different than mentioned in abstract, please check and unify.
- From line 192 to end of discussion: if you considered this issue the effect of season, what about the first part of the discussion as lower than or higher than or similar, did you considered this point, or the literatures considered this point, I see delete this point from discussion as you did not mentioned as well in the aim of the work, OR reconsider this issue in the aim of the wor and rewrite the discussion with other literatures considered this point.
- Line 242: you should mention the MPL of cheese not milk and correct it whenever mentioned in the manuscript as your study on cheese not milk.
- References should be updated till 2023.
- Please add the ethical approval ID.
- Please add missed references in all parts of materials and methods section.
-

Author Response
Reviewer 2:
The manuscript entitled “Incidence and levels of aflatoxin M1 in cheese manufactured in Pernambuco State, Brazil” is interesting as investigated the prevalence and levels of AFM1, as a major dangerous carcinogenic to humans, in artisanal and manufactured coalho and Mozzarella type cheese by using HPLC, but the following comments should be corrected as followed:
Comment 1: The title should be changed as “Incidence and levels of aflatoxin M1 in artisanal and manufactured cheese in Pernambuco State, Brazil”, or “Incidence and levels of aflatoxin M1 in coalho and Mozzarella cheese in Pernambuco State, Brazil”.
Response: Done
Comment 2: The English should be extensively edited in whole the manuscript.
Response: The entire manuscript in the revised form, was exclusively edited for English.
Comment 3: The samples number are too small to be representative for the obtained results, as well as compared with the literatures in the same area of the present work.
Response: Thank you. In this study, a non-probabilistic convenience sampling was conducted, which accounted for approximately 70% of the producers in the study area. It's important to note that not all producers voluntarily agreed to participate in the research. Despite this, the authors believe that the 70% participation rate was substantial, particularly because it was the first study of its kind conducted in the region.
Comment 4: In whole manuscript: change maximum permitted limit to maximum permissible limit.
Response: Done
Comment 5: In the manuscript, HPLC was ignored as an important method of detection and quantification of AFM1, please mention in introduction, result, discussion and conclusion, as well as discuss with the other same used literatures.
Response: Updated.
Comment 6: Line 6: change level to levels.
Response: Done
Comment 7: Line 8: change type to types.
Response: Done
Comment 8: Line 20: remove aflatoxin.
Response: “Aflatoxin” was removed.
Comment 9: Line 21: Add before the word first: is the “This is the first…”.
Response: This was added.
Comment 10: Line 28: change cheese to cheeses.
Response: Done
Comment 11: Line 42: remove the word five.
Response: “five” was removed
Comment 12: Line 65: change higher to high.
Response: Changed
Comment 13: Line 68 and line 69: 0.07µg/kg and 0. 04µg/kg are different than mentioned in Table 1, please check and unify.
Response: This was corrected as per kind suggestion.
Comment 14: Line 72: ab are conflicting, as mentioned when comparing rows and columns, this is not mentioned in this work, please revise the sstistics and only write one letter as well as revise the result and the discussion with this regard to the different types of examined cheeses.
Response: Done
Comment 15: Line 75: the reference no. [19] is too old and since 2011 to 2023 sure several investigations so update the reference with this regard.
Response: The authors concur; however, each country has established maximum permissible limits (MPLs) for mycotoxins in food and food products. Ref. 19 (now Ref. 20) is an official document published by the Brazilian regulatory agency (ANVISA) that outlines the MPLs for mycotoxins in food and food products consumed in Brazil. There are no recent updates from the Brazilian government available at this time.
Comment 16: Below Table 1: Describe the definition of range*.
Response: Thank you, a statement was added as per the recommendation.
Comment 17: The median value mention in line 91 0.041 is wrong it should be 0.04 as well as line 93 the median value mentioned is 0.041 it is wrong and should be 0.040, please revise all values and correct them.
Response: Thank you, the authors have revised these values.
Comment 18: Line 92: the word similar is wrong as they are not similar as the max level was 0.046µg/kg.
Response: Thank you, the authors corrected the mistake in the text.
Comment 19: Line 97: change IC to MC.
Response: Thanks, the mistake was corrected.
Comment 20: Line 99: complete the sentence.
Response: The sentence was completed.
Comment 21: Line 103: remove the word table 1 and rewrite the sentence again.
Response: Removed.
Comment 22: The figure 1: is mismatched with the values mentioned in table1, especially MM, please correct for all values for all types of examined cheeses.
Response: Thank you. After re-confirming the data (text, Table 1, and Figure 1), the results were found to be accurate. Furthermore, the black line inside each box in Figure 1 represents the median.
- Discussion:
Comment 23: You should mention your result first then compare it with others.
Response: Thank you, this issue has been resolved in the revised version of the manuscript.
Comment 24: The collected compared literatures should be exclusively with the examined two types of cheeses: artisanal and manufactured coalho and Mozzarella, please collect more literatures with this regards to make discussion fair and sounds good.
Response: Please refer to the response to Comment 26.
Comment 25: Rewrite the discussion with the previously mentioned regards.s
Response: Please refer to the response to Comment 26.
- Table 2:
Comment 26: The comparison should be exclusively with the examined two types of cheeses: artisanal and manufactured coalho and Mozzarella.
Response: The authors appreciate the reviewer's comment. The cheeses described in the study are only found in the Araripe Sertão and Agreste regions of Pernambuco state, Brazil, and no other studies exist at this time. Therefore, the authors have considered data relevant to any type of cheese (Table 2). Regarding the present manuscript, the study shows that this is the first evaluation of these cheeses, meaning that there are no prior details available to support the results presented. The authors made every effort to collect data before submitting their article, focusing on artisanal and manufactured coalho and mozzarella cheeses. In response to the reviewer's comment, only one study was found and is already cited in the manuscript: "In the north region, the state of Amazonas, Brazil, a study showed coalho, buffalo coalho, mozzarella, buffalo mozzarella, and Minas Frescal cheese (Ref.12)."
Comment 27: In the mentioned unit used ng/Kg and mg/Kg but your compared unit is µg/kg, how come? So you should harmonize between the discussed units to be fair as writing foot note below each unit to the equivalent µg/kg.
Response: The units were updated to µg/kg.
Comment 29: Line 172: 250ng/Kg is different than mentioned in abstract, please check and unify.
Response: This was corrected and updated.
Comment 30: From line 192 to end of discussion: if you considered this issue the effect of season, what about the first part of the discussion as lower than or higher than or similar, did you considered this point, or the literatures considered this point, I see delete this point from discussion as you did not mentioned as well in the aim of the work, OR reconsider this issue in the aim of the work and rewrite the discussion with other literatures considered this point.
Response: However, the authors did not evaluate this issue, but the authors at this point onward merely complemented the data with “seasonality” as an additional factor to be involved. Now taken into account, the "seasonality" as an additional factor could be considered in future studies.
Comment 31: Line 242: you should mention the MPL of cheese not milk and correct it whenever mentioned in the manuscript as your study on cheese not milk.
Response: The authors updated this statement.
Comment 32: References should be updated till 2023.
Response: Please refer to the response to Comment 26.
Comment 33: Please add the ethical approval ID.
Response: “Institutional Review Board Statement: The study was conducted in accordance with the Declaration of Helsinki, and was approved by the Ethics Committee for Human Research at UFRPE (CAAE: 5354.204/2022). Participants completed a Free and Informed Consent Form (FICF).” was added, as per kind suggestion.
Comment 34: Please add missed references in all parts of materials and methods section.
Response: The reference(s) shown in these sections has (have) been updated.

Reviewer 3 Report
Manuscript ID: toxins-2220018
Type of manuscript: Article
Title: Incidence and levels of aflatoxin M1 in cheese manufactured in Pernambuco State, Brazil
Disclaimer: This review was supposed to be a double-blind peer review. However, this is not possible at all, as the authors have quoted themselves. (line 286)
The authors wrote an interesting manuscript about the quantification of aflatoxin M1 in cheese.
The article is written in a very good English and fulfills the requirements of a good scientific work.
In this study, the authors have used an already validated and accepted method: an immunoaffinity column for sample preparation followed by HPLC coupled to a fluorescence detector.
Jager and coworker (DOI:10.1016/j.foodcont.2013.02.016) could already show in the paper of 2013 that Brazilian milk and milk products have higher but still permitted levels of aflatoxins. The number of samples tested was relatively small at that time (milk n=65, milk powder n= 4, cheese n = 10).
In the present manuscript, 14 mozzarella and 14 coalho samples were tested for AFM1. All samples were positive, but none of them had AFM1 levels above the Brazilian maximum permitted limit.
I expected the number of samples tested in this present study to be significantly higher! Twenty eight samples is much too less for such an article.
Assem Elkak et al. (doi:10.1016/j.foodcont.2011.10.033) randomly collected n = 111 cheese samples from local small dairy farms.
Carlos Humberto Corassin et al. (doi:10.3390/dairy3040057) analysed beside 111 milk samples n = 57 cheese samples.
Salman Mohammadi et al (DOI:10.1111/1471-0307.12866) published an systematic review article about Aflatoxin-M1 contamination in west Asian cheese samples: table 2 in this article shows the sample sizes of the reviewed studies - on average over one hundred samples each study.
Even for the current studies cited in table 2, sample sizes larger than 60 are common.
some small comments:
Figure 1: include the number of samples in the figure caption.
Table 2: please convert all AFM1 levels to µg/kg. Dont mix mg/kg and ng/g with µg/kg.
if possible, please add a row beside the "positive", which contains the proportion of samples above the Brazilian maximum permitted limit.
first entry (ref 22): none of the analysed cheese samples were positive. see Table 1 in ref 22.
Chapter Result: please use only one unit for the content of AFM1
line 182: insert a space between number and unit
line 302-304: insert a space between numbers and units
Only after increasing the number of samples examined significantly, I can recommend the acceptance of the manuscript for publication. With this low number of samples, I will recommend to the editor to reject this work.
Author Response
Reviewer 3:
Comment 1: Disclaimer: This review was supposed to be a double-blind peer review. However, this is not possible at all, as the authors have quoted themselves. (line 286)
Response: Sorry, the authors did not get the point raised by reviewer.
Comment 2: The authors wrote an interesting manuscript about the quantification of aflatoxin M1 in cheese.
Response: Thanks.
Comment 3: The article is written in a very good English and fulfills the requirements of a good scientific work.
Response: Thanks.
Comment 4: In this study, the authors have used an already validated and accepted method: an immunoaffinity column for sample preparation followed by HPLC coupled to a fluorescence detector.
Response: Thank you, yes, this method has already been validated by the authors in their Lab.
Comment 5: Jager and coworker (DOI:10.1016/j.foodcont.2013.02.016) could already show in the paper of 2013 that Brazilian milk and milk products have higher but still permitted levels of aflatoxins. The number of samples tested was relatively small at that time (milk n=65, milk powder n= 4, cheese n = 10).
Response: The authors only used the method that was validated in the same research group.
Comment 6: In the present manuscript, 14 mozzarella and 14 coalho samples were tested for AFM1. All samples were positive, but none of them had AFM1 levels above the Brazilian maximum permitted limit.
Response: Yes, none of the samples exceeded the Brazilian maximum permissible limit of 2.5 µg/kg.
Comment 7: I expected the number of samples tested in this present study to be significantly higher! Twenty eight samples is much too less for such an article.
“Assem Elkak et al. (doi:10.1016/j.foodcont.2011.10.033) randomly collected n = 111 cheese samples from local small dairy farms; Carlos Humberto Corassin et al. (doi:10.3390/dairy3040057) analysed beside 111 milk samples n = 57 cheese samples; Salman Mohammadi et al (DOI:10.1111/1471-0307.12866) published an systematic review article about Aflatoxin-M1 contamination in west Asian cheese samples: table 2 in this article shows the sample sizes of the reviewed studies - on average over one hundred samples each study.”
Response: According to the methodology, a non-probabilistic convenience sampling was carried out, which represented approximately 70% of the producers in the study area, considering that not all of them voluntarily agreed to participate in the research. However, the authors believe that this percentage was quite significant, mainly because it is the first study of its kind applied in the region.
Comment 8: Even for the current studies cited in table 2, sample sizes larger than 60 are common.
Response: Please refer to the response to Comment 7.
some small comments:
Comment 9: Figure 1: include the number of samples in the figure caption.
Response: This was updated as suggested.
Comment 10: Table 2: please convert all AFM1 levels to µg/kg. Dont mix mg/kg and ng/g with µg/kg.
Response: All values concerning AFM1 levels mentioned in arbitrary units have been updated to µg/kg.
Comment 11: if possible, please add a row beside the "positive", which contains the proportion of samples above the Brazilian maximum permitted limit.
Response: None of the samples exceeded the Brazilian maximum permissible limit, as indicated in both Table 1 and the accompanying text.
Comment 12: first entry (ref 22): none of the analysed cheese samples were positive. see Table 1 in ref 22.
Response: This was updated from 'NG' to '0' according to Table 1 in Ref. 22 (now Ref. 23).
Comment 13: Chapter Result: please use only one unit for the content of AFM1
Response: The previously mentioned AFM1 levels in "mg/kg, ng/g, µg/L, and/or ng/kg" have been updated and can now be found consistently as "µg/kg" throughout the manuscript wherever random units appeared.
Comment 14: line 182: insert a space between number and unit
Response: This was updated as kindly suggested.
Comment 15: line 302-304: insert a space between numbers and units
Response: Equally updated as kindly suggested.
Comment 16: Only after increasing the number of samples examined significantly, I can recommend the acceptance of the manuscript for publication. With this low number of samples, I will recommend to the editor to reject this work.
Response: Please refer to the response to Comment 7.

Round 2
Reviewer 2 Report
The revised version of the manuscript is fine EXCEPT the number of samples still very low, if the journal do not mind this issue, so it is accepted for publication as it is